# Weather and climate forecasting with neural networks: using GCMs with different complexity as study-ground

Sebastian Scher[1] and Gabriele Messori[1,2]

[1]Department of Meteorology and Bolin Centre for Climate Research, Stockholm University, Stockholm, Sweden
[2]Department of Earth Sciences, Uppsala University, Uppsala, Sweden

**Correspondence:** Sebastian Scher sebastian.scher@misu.su.se

**Abstract.** Recently, there has been growing interest in the possibility of using neural networks for both weather forecasting and the generation of climate datasets. We use a bottom-up approach for assessing whether it should, in principle, be possible to do this. We use the relatively simple General Circulation Models (GCMs) PUMA and PLASIM as a simplified reality on which we train deep neural networks, which we then use for predicting the model weather at lead times of a few days. We specifically assess how the complexity of the climate model affects the neural network's forecast skill, and how dependent the skill is on the length of the provided training period. Additionally, we show that using the neural networks to reproduce the climate of general circulation models including a seasonal cycle remains challenging – in contrast to earlier promising results on a model without seasonal cycle.

## 1  Introduction

Synoptic weather forecasting (forecasting the weather at lead times of a few days up to 2 weeks), has for decades been dominated by computer models based on physical equations – the so called Numerical Weather Predictions (NWP) models. The quality of NWP forecasts has been steadily increasing since their inception (Bauer et al., 2015), and these models remain the backbone of virtually all weather forecasts. However, the fundamental nature of the weather forecasting problem can be summarized as: "starting from today's state of the atmosphere, we want to predict the state of the atmosphere $x$ days in the future". Thus posed, the problem is a good candidate for supervised machine learning. While long thought unfeasible, the recent success of machine learning techniques in highly complex fields such as image and speech recognition warrants a review of this possibility. Machine learning techniques have already been used to improve certain components of NWP and climate models – mainly parameterization schemes (Krasnopolsky and Fox-Rabinovitz (2006); Rasp et al. (2018); Krasnopolsky et al. (2013); O'Gorman and Dwyer (2018)), to aid real-time decision making McGovern et al. (2017), to exploit observations and targeted high-resolution simulations to enhance earth system models (Schneider et al., 2017), for El-Niño predictions (Noote-

boom et al., 2018) and to predict weather forecast uncertainty (Scher and Messori, 2018).

Recently, in addition to using machine-learning to enhance numerical models, there have been ambitions to use it to tackle weather forecasting itself. The holy grail is to use machine-learning, and especially "deep learning", to completely replace NWP models, although opinions may diverge on if and when this will happen. Additionally, it is an appealing idea to use neural networks/deep learning to emulate very expensive General Circulation Models (GCMs) for climate research. Both these ideas have been tested with some success for simplified realities (Dueben and Bauer, 2018; Scher, 2018). In Scher (2018), a neural network approach was used to skillfully forecast the "weather" of a simplified climate model, as well as emulate its climate. Dueben and Bauer (2018), based on their success in forecasting reanalysis data regridded to very low resolution, concluded that it is "fundamentally" possible to produce deep-learning based weather forecasts.

Here, we build upon the approach from Scher and apply it to a range of climate models with different complexity. We do this in order to assess: 1) how the skill of the neural network weather forecasts depends on the available amount of training data; 2) how this skill depends on the complexity of the climate models; and 3) under which conditions it may be possible to make stable climate simulations with the trained networks, and how this depends on the amount of available training data. (1) is of special interest for the idea of using historical observations in order to train a neural network for weather forecasting. As the length of historical observations is strongly constrained (~ 100 years for long renalyses assimilating only surface observations, and ~40 years for reanalyses assimilating satellite data), it is crucial to assess how many years of past data one would need to produce meaningful forecasts. The value of (2) lies in evaluating the feasibility of climate models as a "simplified reality" for studying weather forecasting with neural networks. Finally, (3) is of interest when one wants to use neural networks not only for weather forecasting, but for the distinct yet related problem of seasonal and longer forecasts, up to climate projections.

To avoid confusion, we use the following naming conventions throughout the paper: "model" always refers to physical models (i.e. the climate models used in this study), and will never refer to a "machine learning model". The neural networks are referred to by the term "network".

## 2 Methods

### 2.1 Climate models

To create long climate model runs of different complexity, we used the Planet Simulator (PLASIM) intermediate complexity GCM, and its dry dynamical core: the Portable University Model of the Atmosphere (PUMA) (Fraedrich et al., 2005). Each model was run for 830 years with two different horizontal resolutions (T21 and T42, corresponding to ~5.65 and ~2.8 degrees latitude, respectively) and 10 vertical levels. The first 30 years of each run were discarded as spin-up, leaving 800 years of daily data for each model. The runs will from now on be referred to as PLASIMT21, PLASIMT42, PUMAT21 and PUMAT42. All the model runs produce a stable climate without drift (fig. S1-S10 in the Supplement). Additionally, we regridded PLASIMT42

and PUMAT42 (with bi-linear interpolation) to a resolution of T21. These will be referred to as PLASIMT42_regridT21 and PUMAT42_regridT21.

The PUMA runs do not include ocean and orography and use Newtonian cooling for diabatic heating/cooling. The PLASIM runs include orography, but no ocean-model. The main difference between PLASIM and standard GCMs is that sub-systems of the Earth system other than the atmosphere (e.g. the ocean, sea-ice and soil) are reduced to systems of low complexity. We used the default integration timestep for all four model setups, namely 30 min for PLASIMT42, 20min for PLASIMT21, and 60 min for PUMAT21 and PUMAT42. We regrid all fields to regular lat-lon grids on pressure levels for analysis. An additional run was made with PUMA at resolution T21, but with the seasonal cycle switched off (eternal boreal winter). This is the same configuration as used in Scher (2018) and will be referred to as PUMAT21_noseas.

In order to contextualise our results relative to previous studies, we also run a brief trial of our networks on ERA5 (C3S, 2017) reanalysis data (Section 3.5 , similar to Dueben and Bauer (2018).

## 2.2 Complexity

Ranking climate models according to their "complexity" is a non-trivial task, as it is very hard to define what complexity actually means in this context. We note that here we use the term loosely, and do not refer to any of the very precise definitions of "complexity" that exists in various scientific fields (e.g. Johnson (2009)). Intuitively, one might simply rank the models according to their horizontal and vertical resolutions and the number of physical processes they include. However, it is not clear which effects would be more important (e.g. is a model with higher resolution but less components/processes more or less complex than a low-resolution model with a larger number of processes?). Additionally, more physical processes do not necessarily imply a more complex *output*. For example, very simple models like the Lorenz63 model (Lorenz, 1963) display chaotic behaviour, yet it is certainly possible to design a model with more parameters and a deterministic behaviour.

To circumvent this conundrum, we adopt a very pragmatic approach based solely on the output of the models, and grounded in dynamical systems theory. We quantify model complexity in terms of the the local dimension $d$: a measure of the number of degrees of freedom of a system active locally around a given instantaneous state. In our case, this means that we can compute a value of $d$ for every timestep in a given model simulation. While not a measure of complexity in the strict mathematical or computational senses of the term, $d$ provides an objective indication of a system's dynamics around a given state and, when averaged over a long timeseries, of the system's average attractor dimension. An example of how $d$ may be computed for climate fields is provided in Faranda et al. (2017), while for a more formal discussion and derivation of $d$ we point the reader to Appendix A in Faranda et al. (2019). The approach is very flexible, and may be applied to individual variables of a system (which represent projections of the full phase-space dynamics onto specific sub-spaces, called Poincaré sections), multiple variables or, with adequate computational resources, to the whole available dataset. The exact algorithm used here is outlined in Appendix A1.

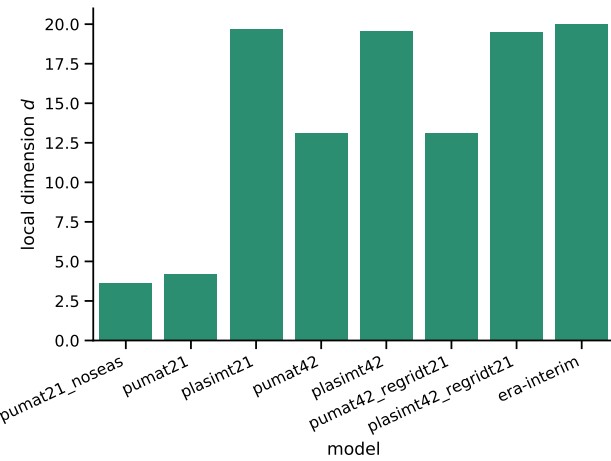

**Figure 1.** Averages of the local dimension $d$ (here used as a data-based measure of "complexity") for the 500hPa geopotential height of the models used in this study and of the ERA-Interim reanalysis.

The local dimension was computed for 38 years of each model run, as well as for the ERA-Interim reanalysis on a 1x1 degree grid over 1979-2016 (Dee et al., 2011). The choice of 38 years was made because this is the amount of available years in ERA-Interim, and the length of the dataseries can affect the estimate of $d$ (Buschow and Friederichs, 2018). Figure 1 shows the results for 500hPa geopotential height. The complexity of PUMA increases with increasing resolution, whereas both the low and the high resolution PLASIM model have a complexity approaching that of ERA-Interim. Thus – at least by this measure – they are comparable to the real atmosphere. The high resolution runs regridded to T21 have nearly the same complexity as the original high resolution runs. The ranking is the same for nearly all variables and levels (fig. S11 in Supplement). For the rest of the paper, the term "complexity" or "complex" always refers to the local dimension $d$.

## 2.3 Neural networks

Neural networks are in principle a series of non-linear functions with weights determined through training on data. Before the training, one has to decide the *architecture* of the network. Here, we use the architecture proposed by Scher (2018), which is a convolutional decoder-encoder, taking as input 3d model fields and outputting 3d model fields of exactly the same dimension. It was designed and tuned in order to work well on PUMAT21 without seasonality (for details see Scher (2018)). In order to ease comparison with previous results, in the main part of this study we use the same network layout and hyperparameters as in (Scher, 2018), except for the number of epochs (iterations over the training set) the network is trained over. In the original configuration only 10 epochs were used. It turned out that, especially for the more complex models, the training was not saturated after 10 epochs. Therefore, here we train until the skill on the validation data has not increased for 5 epochs, with a maximum of 100 epochs. The layout is depicted in fig. 2. The possibility of re-tuning the network and its implications are discussed in section 3.4. For the networks targeted to create climate simulations (hereafter called climate-networks) we deviate from this

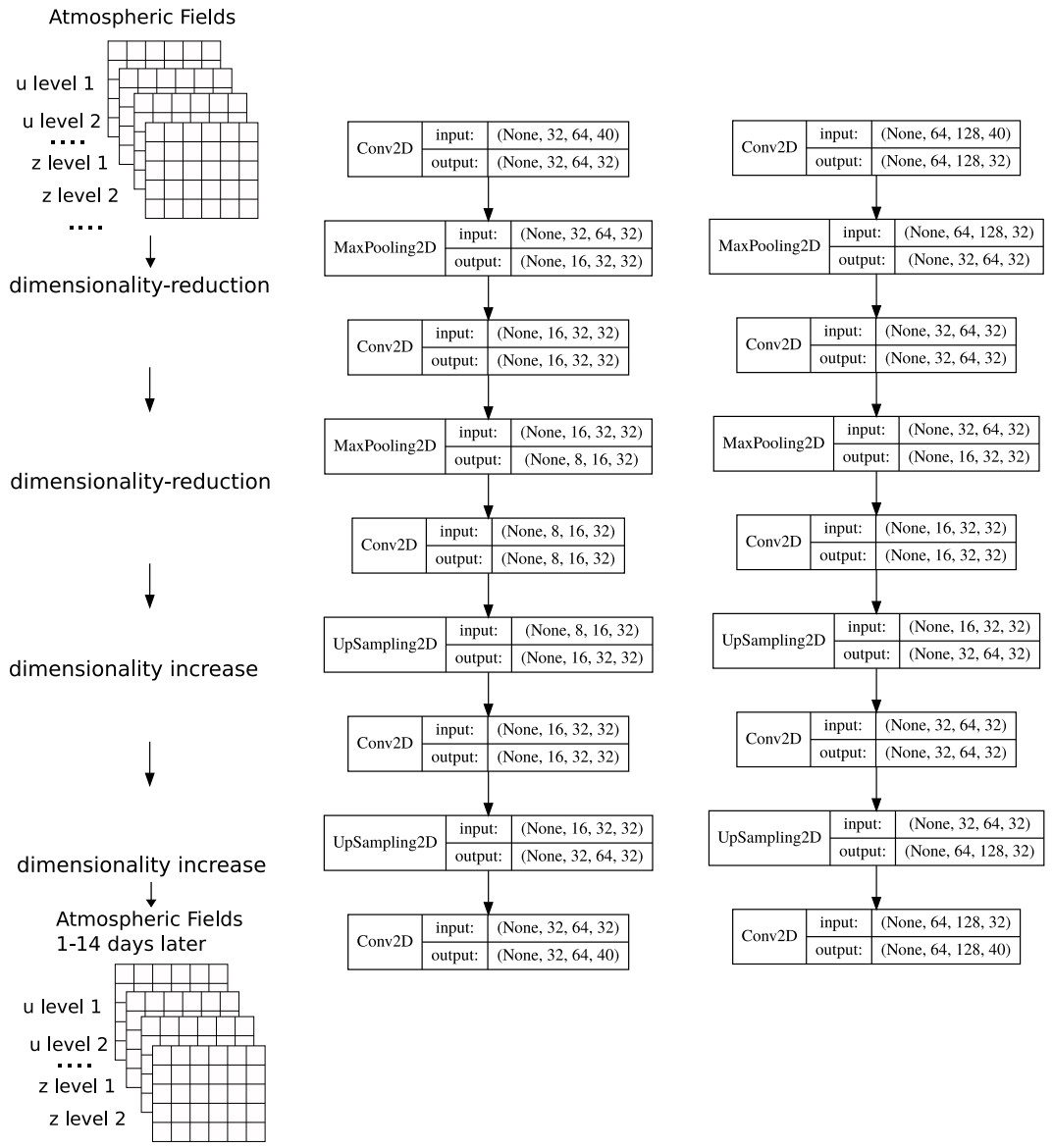

**Figure 2.** Architecture of the neural network for the models with resolution T21 (left) and T42 (right). Each box describes a network layer, with the numbers indicating the dimension of (None, lat, lon, level). "None" refers to the time-dimension that is not relevant here, but which we include in the schematic since it is part of the basic notation of the used software library. Figure based on Fig. S1 from Scher (2018).

setup: here, we include the day of year as additional input to the network, in the form of a separate input channel. To remain consistent with the encoder-decoder setup, the output also contains the layer with the day of year. However, when producing the network climate runs, the predicted day of the year is discarded.

The last 10% samples of the training data are used for validation. This allows to monitor the training progress, control over-fitting (the situation where the network works very well on the training data, but very poorly on the test data), and potentially limit the maximum number of training epochs. As input to the neural networks we use 4 variables (u, v, t and z) at 10 pressure levels; each variable at each level is represented as a separate input layer (channel). These 4 variables represent the full state of the PUMA model. PLASIM has 3 additional atmospheric variables related to the hydrological cycle (relative humidity, cloud liquid water content and cloud cover). In order to keep the architecture the same, these are not included in the standard training, and only used for a test in section 3.5.

All networks are trained to make 1-day forecasts. Longer forecasts are made by iteratively feeding back the forecast into the network. We did not train the network directly on longer lead-times, based on the finding of Dueben and Bauer (2018) that it is easier to make multiple short forecasts compared to a single long one. Due to the availability of model data and in keeping with Scher (2018), we chose 1-day forecasts as opposed to the shorter forecast step (1 hour) in Dueben and Bauer (2018). For each model, the network was trained with a set of 1, 2, 5, 10, 20, 50, 100, 200, 400 and 800 years. Since with little training data the network is less constrained, and the training success might strongly depend on exactly which short period out of the model run is chosen, the training for periods up to and including 20 years were repeated 4 times, shifting the start of the training data by 10, 20, 30 and 40 years. The impact of the exact choice of training period is discussed where appropriate. All the analyses shown in this paper are performed on the forecasts made on the first 30 years of the model run, which were never used during training and therefore provide objective scores (the 'test' dataset).

## 2.4 Metrics

All network forecasts are validated against the underlying model the network was trained on (e.g. for the forecasts of the network trained on PLASIMT21 the "truth" is the evolution of the PLASIMT21 run). We specifically adopt two widely used forecast verification metrics, namely the Root Mean Square Error (RMSE) and the Anomaly Correlation Coefficient (ACC). The RMSE is defined as:

$$RMSE = \sqrt{\overline{(prediction - truth)^2}} \tag{1}$$

where the overbar denotes a mean over space and time (for global measures), or over time only (for single gridpoints). The ACC measures the spatial correlation of the forecast anomaly fields with the true anomaly fields for a single forecast. The anomalies are computed with respect to a 30-day running climatology, computed on 30 years of model data (similar to how the European Centre for Medium Range Weather Forecasts computes its scores for forecast validation).

$$ACC_t = correlation$$

$$([truth_{1,1} - clim_{1,1}, ....., truth_{nlat,nlon} - clim_{nlat,nlon}], [prediction_{1,1} - clim_{1,1}, ....., prediction_{nlat,nlon} - clim_{nlat,nlon}])$$

$$(2)$$

To compute a score over the whole period, the ACC for all individual forecasts are simply averaged:

$$ACC = \overline{[ACC_1, ...., ACC_{nforecast}[} \tag{3}$$

The ACC ranges from -1 to 1, with 1 indicating a perfect correlation of the anomaly fields, and 0 no correlation at all.

For the evaluation of the brief ERA5-test in section 3.6, we use the mean absolute error instead of RMSE to keep in line with Dueben and Bauer (2018). The mean absolute error is defined as

$$MAE = \overline{|prediction - truth|} \tag{4}$$

## 3    Results

### 3.1    Forecast skill in a hierarchy of models

We start by analyzing the RMSE and ACC of two of the most important variables: 500hPa gepotential height (hereafter zg500) and 800hPa temperature (hereafter ta800). The first is one of the most commonly used validation metrics in NWP; the second

is very close to temperature at 850hPa, which is another commonly used validation metric. We focus on the networks trained with 100 years of training data, which is the same length as in Scher (2018), and is of special interest because it is roughly the same length as is available in current century-long reanalyses like ERA-20C and the NCEP/NCAR 20CR (Compo et al., 2011; Poli et al., 2016). Figure 3 shows the global mean RMSE of network forecasts at lead times of up to 14 days for all models for both zg500 and ta800. Additionally, the skill of persistence forecasts (using the initial state as forecast) is shown with dashed

lines. As expected, the skill of the network forecasts decreases monotonically with lead-time. Unsurprisingly, the network for PUMAT21_noseas – the least complex model – has the highest skill (lowest error) for all lead-times for both variables, followed by the network for PUMAT21. The network for PUMAT42, which is more complex than PUMAT21, but less complex than the two PLASIM runs, lies in between. At a lead time of 1 day, the networks for both PLASIM runs have very similar skill, but the PLASIMT42 network has higher errors at longer lead-times, despite their very similar complexity. Here we have to note

that at long lead-times and near-zero ACC, RMSE can be hard to interpret since it can be strongly influenced by biases in the forecasts. When looking at the ACC instead (higher values better), the picture is very similar. The network forecasts outperform the persistence baseline (dashed lines) at nearly all lead-times, except for the PLASIMT21 and PLASIMT42 cases, where the RMSE of the network forecasts is higher from around 9 days on (depending on the variable). The periodic behaviour of the

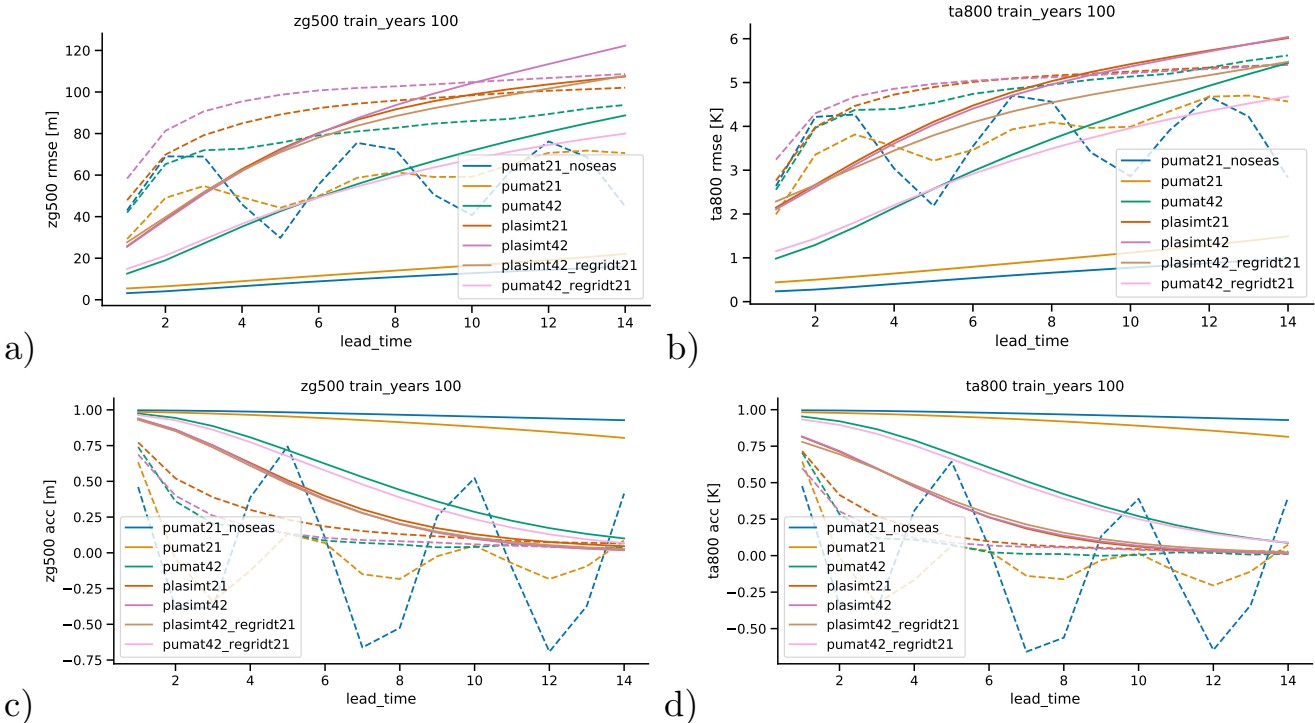

**Figure 3.** Root Mean Square Error (RMSE) and Anomaly Correlation Coefficient (ACC) of 500hPa gepotential height (a,c) and 800hPa temperature (b,d) for network forecasts (solid lines) and persistence forecasts (dashed lines) for all models for different lead times (in days). All forecasts are based on 100 years of training data.

persistence forecast skill for PUMA is caused by east-wards travelling rossby-waves, whose structure is relatively simple in the PUMA model. For the T42 runs that were regridded to T21 before the training the results are as follows: for PUMA, the skill of the network in predicting the regridded version of the T42 is very similar to the skill on the original T42 run. For PLASIM the skill on the regridded T42 run is comparable to both the skill on T42 and on T21 runs, albeit closer to the latter. Indeed,

5    the skills on the original PLASIM T42 and T21 runs are much closer to each other than for PUMA. Regridding the network predictions of the two T42 runs to the T21 grid thus results in only very small changes relative to the difference between the models, especially at longer lead-times (not shown).

We next turn our attention to the spatial characteristics of the forecast error. Figure 4 shows geographical plots of the RMSE

10    for 6-day forecasts of the networks trained with 100 years of data (the same training length as in fig. 3). In agreement with the global mean RMSE analyzed before, the network for PUMAT21_noseas has lowest errors everywhere (fig. S12 in Supplement), followed by the network for PUMAT21. The networks for PLASIMT21 and PLASIMT42 have a more complicated spatial error structure, and the mid-latitude storm-tracks emerge clearly as high-error regions. The zonally non-uniform distribution is likely

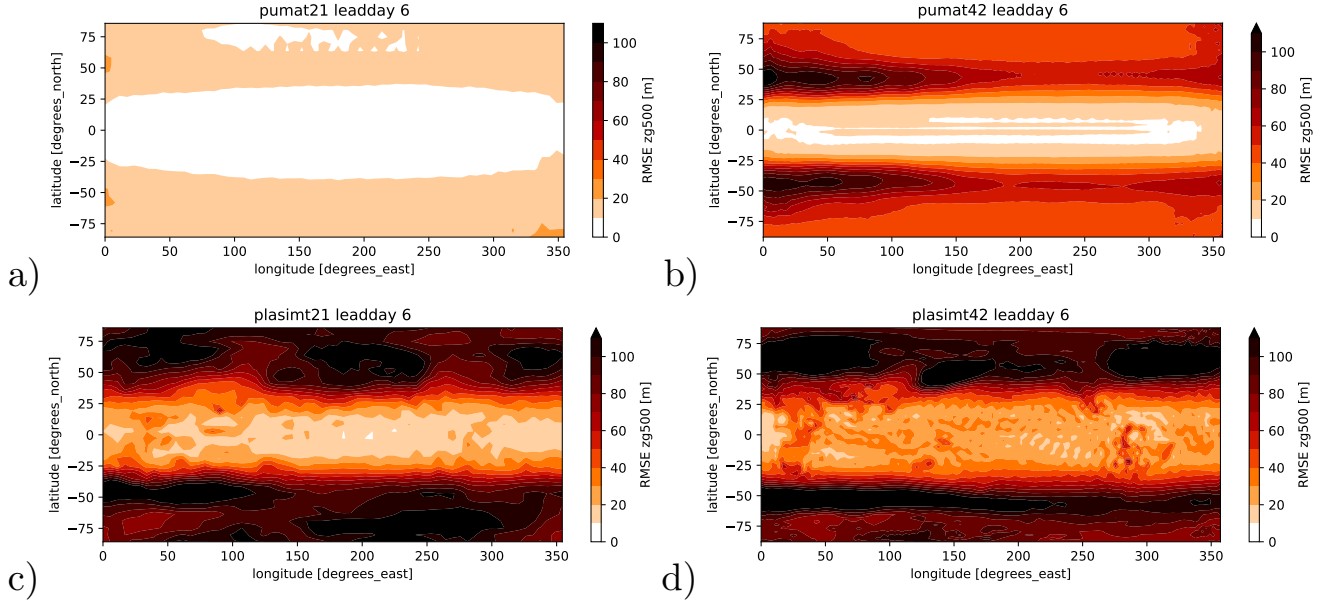

**Figure 4.** Maps of RMSE of the 6-day network forecasts, for the networks trained with 100 years for PUMAT21 (a), PUMAT42 (b), PLASIMT21 (c) and PLASIMT42 (d).

caused by the influence of orography (present in the PLASIM runs but not in the PUMA runs). At lead-time 1 day, the errors of the network forecasts for PUMAT42 are nearly symmetric (not shown), but at longer lead-times a zonal asymmetry emerges (fig. 4b). This is probably related to the fact that the neural network used here does not wrap around the boundaries.

### 3.2 Dependence of forecast skill on amount of training years

A key issue is the extent to which the above results, and more generally the skill of the network forecasts, depend on the length of the training period used. Fig. 5 shows the skill of the network forecasts for 500hPa geopotential height for different training lengths, for a lead-time of 1 day (a,c) and 6 days (b,d). As mentioned in the methods section, the networks with short training periods were trained several times with different samples from the model runs. The shading in the figure represents the uncertainty of the mean for these multi-sample networks, which is negligibly small. For the 1-day forecasts (fig. 5a, c), the

results are as expected: the skill increases with an increasing number of training years, both in terms of RMSE and ACC. This increase is strongly nonlinear and, beyond ~100 years, the skill benefit of increasing the length of the training set is limited. This suggests that the complete model-space is already encompassed by around 100 years of daily data. More years will not provide new information to the network. However, it might also be the case that there is in fact more information in more years, but that the network is not able to utilize this additional information effectively. For the 6-day forecasts (fig. 5b, d), the

PLASIMT21 networks display a counter-intuitive behaviour: the skill (in terms of RMSE) does not increase monotonically with increasing length of the training period, but decreases from 100 years to 200 years, while for >200 years it increases

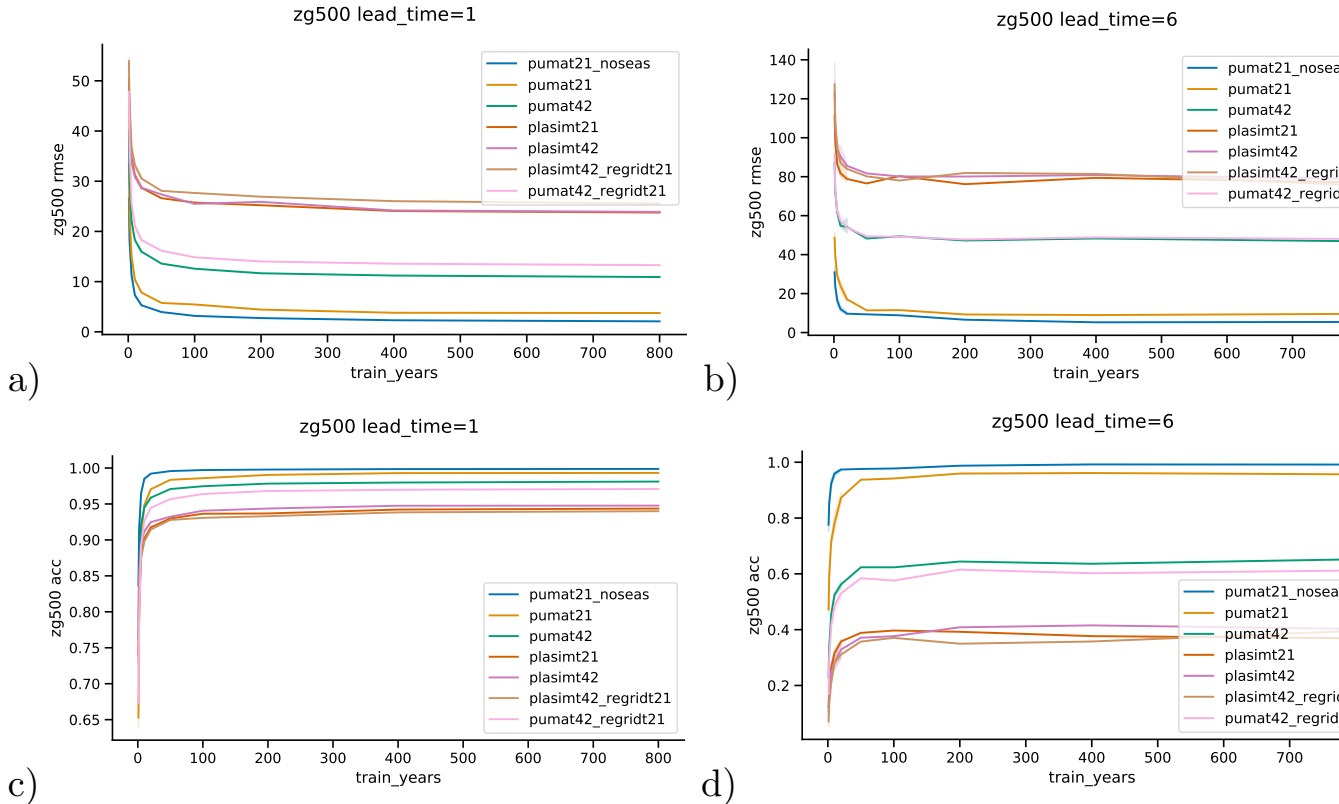

**Figure 5.** Dependence of network forecast skill on the length of the training period. Shown is the Root Mean Square Error (RMSE) and Anomaly Correlation Coefficient (ACC) of the network-forecasted 500hPa gepotential height for networks trained on different amounts of training years. Each line represents one model. The shading on the left side of the plots represents the 5-95 uncertainty range of the mean RMSE/ACC, estimated over networks with different training samples.

again. A similar result – albeit less pronounced – is also seen for the PLASIMT42_regridT21 networks, and also – in a slightly different form – for the skill measured via the ACC. To interpret this, one has to remember that the networks are all trained on 1-day forecasts. For 1-day forecasts the skill is indeed increasing with increasing training length, and the PUMAT21 network trained on 200 years makes better forecasts than the one trained on 100 years (fig. 5a, c). Intuitively one would assume this to translate to increased skill of the consecutive forecasts used to create the 6 day forecasts. The fact that this is not the case here might be caused by non-linear error growth. Some networks might produce slightly lower errors at lead-day 1, but those particular errors could be faster-growing than those of a network with larger day-1 errors.

### 3.3 Climate runs with the networks

The trained networks are not limited to forecasting the model "weather", but can also be used to generate a climate run starting from an arbitrary state of the climate model. For this, we use the climate networks that also include the day of year as input (see

methods section). Of special interest is the question of whether the climate time-series obtained from the network is stable. In Scher (2018), the network climate for PUMAT21_noseas (using 100 years of training data) was stable and produced reasonable statistics compared to the climate model. We trained our climate networks on both 30 years and 100 years of data for all models with seasonal cycle. While the networks were all stable, they do not produce particularly realistic representations of the model

climates. After some time, the storm tracks look unrealistic, and the seasonal cycle is also poorly represented (e.g. in some years, some seasons are skipped – see videos S1-S8 in the supplement that show the evolution of zg500 and zonal windspeed at 300hPa for the network climate runs). Figure 6 a) shows the evolution of ta800 at a single gridpoint at 76 °N for PUMAT21 and the climate network trained on 30 years, started from the same (randomly selected) initial state (results for different randomly selected initial states are very similar, not shown). The network is stable, but the variance is too high and some summers are

left out. Surprisingly, the network for 30 years of PLASIMT21 (fig. 6 b) produced a more realistic climate. Training on 100 years instead of 30 years does not necessarily improve the quality of the climate-runs (fig. 6 c,d). For PLASIMT21, in fact, the network climate trained on 100 years is the worse performer. Interestingly, the mean climate of the PLASIMT21 network is reasonably realistic for the network trained on 100 years (fig. 7), and partly also for the network trained on 30 years (not shown), whereas the mean climates for the PLASIMT42 and PUMAT42 networks have large biases (see fig S11-17 in the

Supplement).

All our networks were trained to make 1-day forecasts. The influence of the seasonal cycle on atmospheric dynamics – and especially the influence of diabatic effects – may be very small for 1-day predictions. This could make it hard for the network to learn the influence of seasonality. To test this, we repeated the training of the climate network for PLASIMT21, but now

training on 5-day forecasts. This improved the seasonality of ta800 at our example gridpoint (fig. S19 in the supplement), but the spatial fields of the climate run still become unrealistic (video S9 in the supplement).

### 3.4 Impact of re-tuning

The design of this study was to use an already established neural network architecture – namely one tuned on a very simple model – and apply it to more complex models. However, it is of interest to know how much tuning the network architec-

ture to the more complex models might increase forecast skill. Therefore, the same tuning procedure as in Scher (2018) for PUMAT21_noseas was repeated for PLASIMT21. Surprisingly, the resulting configuration for PLASIMT21 was exactly the same as for for PUMAT21_noseas. Thus, even with re-tuning the results would be the same. As a caveat, we note that tuning neural networks is an intricate process, and many arbitrary choices have to be made. Notably, one has to pre-define the different configurations that are tried out in the tuning (the "tuning space"). It is possible that with a different tuning space for

PLASIMT21, a different configuration would be chosen than for PUMAT21_noseas. However, at least within the tuning space we used, we can conclude that a setup working well for a very simple model (PUMAT21_noseas) is also a good choice for a more complex model like PLASIMT21.

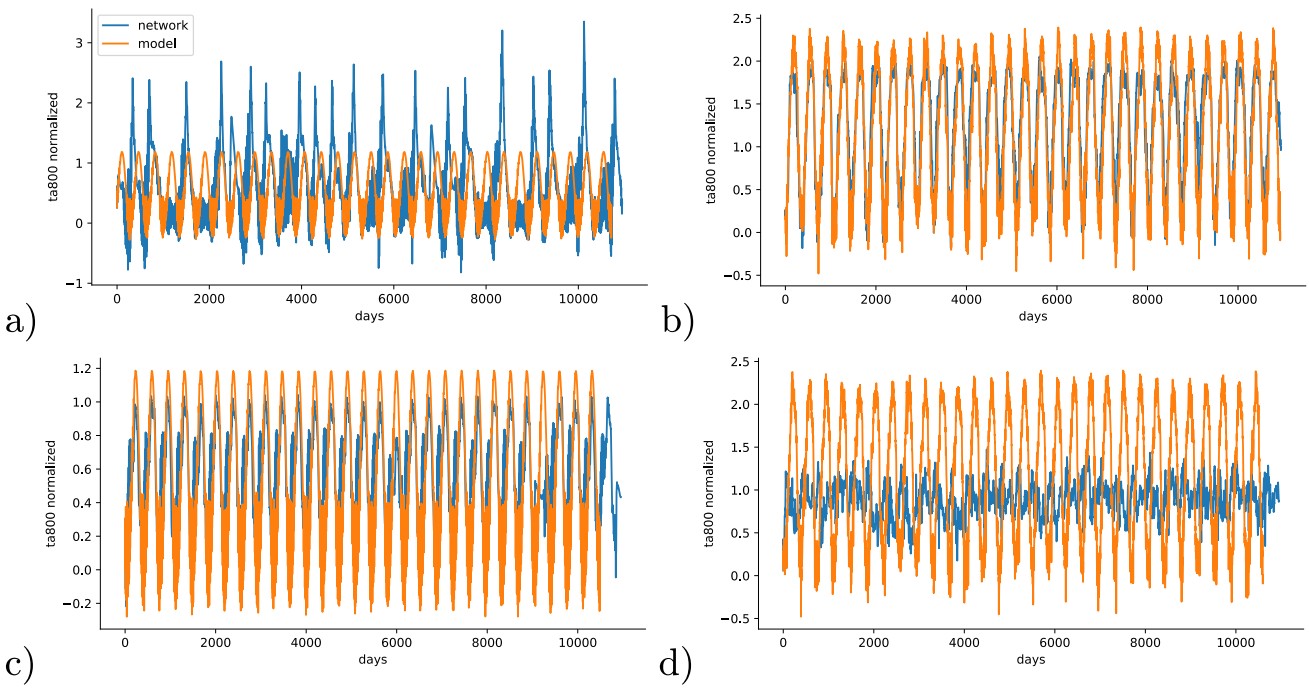

**Figure 6.** Evolution of daily ta800 at a single grid-point at 76°N in the GCM (orange) and in the climate network trained on the GCM (blue), started from the same initial state. The networks were trained on: 30 years of PUMAT21 (a), 30 years of PLASIMT21 (b), 100 years of PUMAT21 (c) and 100 years of PLASIMT21 (d).

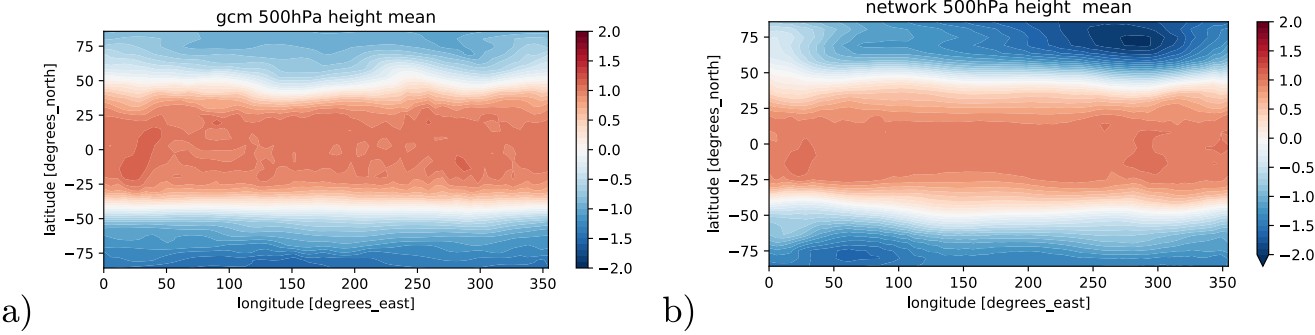

**Figure 7.** 30 year mean of normalized 500hPa geopotential height for PLASIMT21 (a) and the network trained on 100 years of PLASIMT21 (b).

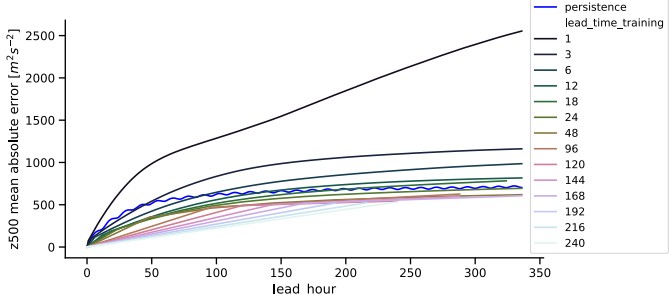

**Figure 8.** Mean absolute error of 500 hPa geopotential height, for networks trained on coarse-grained (T21) ERA5 data. The training was performed on different lead-times. The x-axis denotes the lead-time of the forecast in hours, the colors the lead-time used for training (in hours). Also shown is the persistence forecast (thick blue line).

### 3.5 Impact of including hydrological cycle as input

PLASIM, in contrast to PUMA, includes a hydrological cycle. This is represented by 3 additional 3d state variables in the model (relative humidity, cloud liquid water content and cloud cover). To the test the impact of including these variables in the neural networks, we retrained the network for 100 years of PLASIMT21 including these 3 variables (on 10 levels) as additional

5 channels both in the input and output of the network. The network thus has 70 instead of 40 in- and output channels. Quite surprisingly, including the hydrological cycle variables slightly deteriorated the forecasts of zg500 and ta800, both in terms of RMSE and ACC, except for the ACC of ta800 from lead-times of 6 days onwards (fig. S20 in the supplement).

### 3.6 Performance on reanalysis data

In order to put our results in context, we also trained our network architecture on coarse-grained (regridded to T21) ERA5 (C3S,

2017) reanalysis data. We use exactly the same dataset that Dueben and Bauer (2018) used, namely 7 years (2010-2016) for training, and 8 months for evaluation (January 2017- August 2017). ERA5 is available in hourly intervals, allowing a thorough investigation of which timestep is best for training. As in Dueben and Bauer (2018), we only use zg500, and therefore deviate from the setup used in the rest of this study. We do this in order to be able to directly compare the results in the two studies. We trained networks on lead-times ranging from 1 to 240 hours, and computed the skill of the zg500 forecasts. The results are

shown in fig. 8. The blue line shows the error of persistence forecasts. The network trained on 1h-forecasts is not stable and has very high forecast errors, in contrast to Dueben and Bauer who only obtained good forecasts with training on 1h lead-time. The network trained with 24h lead-time has comparable skill to the network used by Dueben and Bauer (2018) (see their fig. 3a). For longer lead-times, the networks trained on longer lead-times seem to work better. In general, it seems that if one wishes to have forecasts for a certain lead-time, it is best to train directly on that lead-time, at least for mean absolute error.

## 4   Discussion and Conclusions

We have tested the use of neural networks for forecasting the 'weather' in a range of simple climate models with different complexity. For this we have used a deep convolutional encoder-decoder architecture that Scher (2018) developed for a very simple general circulation model without seasonal cycle. The network is trained on the model in order to forecast the model
state 1 day ahead. This process is then iterated to obtain forecasts at longer lead times. We also performed "climate" runs, where the network is started with a random initial state from the climate model run, and then creates a run of daily fields for several decades.

**Potential improvements in the neural network architecture**

In the architecture used in this paper, lat-lon grids are used to represent the fields. The convolution layers consists of a set of
fixed filters that are moved across the domain. Therefore, the same filter is used close to the poles and close to the equator, even though the area represented by a single gridpoint is not the same close to the poles and close to the equator since the Earth is a globe. Using spherical convolution layers as proposed in Coors et al. (2018) would tackle this problem and may lead to improved forecasts and/or more realistic long-term simulations. The tuning of the network architecture used here tuned the depth of the convolution layers, but not the actual number of convolution layers. Therefore, it would be interesting to
explore whether deeper networks (more convolution layers) could improve the performance of the networks. Other possible improvements would be to:

- Include one or more locally connected layers in addition to the convolution layers. While they can increase problems related with overfitting, locally connected layers could learn "local dynamics".

- Include spatial dependence in height. In our architecture, all variables at all heights are treated as individual channels.
One could for example group variables at one height together, or use 3d-convolution.

Our results further support the idea – already proposed by Dueben and Bauer (2018) and Scher (2018) – of training a neural network on the large amount of existing climate model simulations, feeding the trained network with today's analysis of a NWP model and using the network to make a weather forecast. The high computational efficiency of such neural network forecasts would open up new possibilities, for example of creating ensemble forecasts with much large ensemble sizes than the
ones available today. Therefore, this approach would provide an interesting addition to current weather forecasting practice, and also a good example of exploiting the large amount of existing climate model data for new purposes.

**Networks for weather forecasts and climate runs**

One of the aims of this study was to assess whether it is possible to use a simplified reality – in this case a very simple GCM without seasonal cycle – to develop a method that also works on more complex GCMs. We showed that, for the problem of
forecasting the model 'weather', this seems to be the case: the network architecture developed by Scher (2018) also worked on the more complex models used here, albeit with lower skill. The latter point is hardly surprising, as one would expect

the time-evolution of the more complex models to be harder to predict. The neural network forecast outperformed a simple persistence forecast baseline at most lead-times also for the more complex models. The fact that we can successfully forecast the weather in a range of simple GCMs a couple of days ahead is an encouraging result for the idea of weather forecasting with neural networks. We also tried to re-tune the network architecture from Scher (2018) to one of our more complex

models. Surprisingly, the best network configuration that came out of the tuning procedure was exactly the same as the one obtained for the simpler model in Scher (2018). This further supports the idea that methods developed on simpler models may be fruitfully applied in more complex settings. Additionally, we tested the network architecture on coarse-grained reanalysis data, where its skill is comparable to the method proposed by Dueben and Bauer (2018). We also found that, in contrast to the findings of Dueben and Bauer (2018), it seems possible to make valid neural network forecasts of atmospheric dynamics

using a wide range of timesteps (also much longer than 1 hour). This discrepancy might be caused by our use of convolutional layers, in contrast to the local deep neural network that Dueben and Bauer used. A stack of convolution layers may be interpreted as multiple layers, where each layer could possibly make a short-term forecast, and the whole stack a long-term forecast.

The second problem we addressed was using the trained networks to create "climate" runs. Scher (2018) found this generated

a stable climate for the simplest model considered here, which does not have a seasonal cycle. Here, we find that this is to some extent also possible for more complex models. However, even when training on relatively long periods (100 years), the climates produced by the networks have some unrealistic properties, such as a poor seasonal cycle, significant biases in the long-term mean values and often unrealistic storm tracks. The fact that these problems don't occur for the simplest GCM without seasonal cycle, but do occur for the same GCM with seasonal cycle, indicates that seasonality considerably complicates the problem.

While not a solution for creating climate runs, this suggests that for the weather forecasting problem it might be interesting to train separate networks for different times of the year (e.g. one for each month).

*Code and data availability.* The code developed and used for this study is available in the accompanying repository at http://doi.org/10.5281/zenodo.2572863. All external libraries used here are open source. The trained networks and the data underlying the all plots are available in the repository. The model runs can be recreated with the control files (available in the repository) and the source code of PUMA/PLASIM,

which is freely available at https://www.mi.uni-hamburg.de/en/arbeitsgruppen/theoretische-meteorologie/modelle/PLASIM.html ERA-Interim data is freely avialable at https://apps.ecmwf.int/datasets/data/interim-full-daily/levtype=pl/. ERA5-data is freely availableat https://climate.copernicus.eu/climate-reanalysis

## Appendix A

### A1 Computation of the local dimension

Here, we outline very briefly how the local dimension $d$ is computed. To foster easy reproducibility, we present the computation in an algorithm-like fashion, as opposed to formal mathematical notation. For a more rigorous theoretical explanation the

reader is referred to Faranda et al. (2019). The code is available in the repository accompanying this paper (see Code and data availability).

First, we define the distance between the 2-D atmospheric fields at times $t_1$ and $t_2$ as:

$$dist_{t_1,t_2} = sqrt\left(\left(x_{t_1,1} - x_{t_2,1}\right)^2 + \left(x_{t_1,2} - x_{t_2,2}\right)^2 + ... + \left(x_{t_1,N_j} - x_{t_2,N_j}\right)^2\right) \tag{A1}$$

where $j$ is the linear gridpoint index and $N_j$ is the total number of gridpoints. To compute $d_t$, namely the local dimension of a field at time t, we first take the negative natural logarithm of the distances between $t$ and all other timesteps $t_i$ (i.e. all times before and after $t$):

$$g_{t,t_i} = -ln\left(dist_{t,t_i}\right) \tag{A2}$$

and then retain only the distances that are above the 98th percentile of $g_{t,t_i}$:

$$exceedances = g_{t,t_i} - 98^{th}percentile(g_{t,t_i}) \,\forall\, g_{t,t_i} > 98^{th}percentile(g_{t,t_i}) \tag{A3}$$

These are effectively logarithmic returns in phase space, corresponding to cases where the field $x_{t_i}$ is very close to the field $x_t$. According to the Freitas-Freitas-Todd theorem (Freitas et al., 2010), modified in (Lucarini et al., 2012), the probability of such logarithmic returns (in the limit of an infinitely long timeseries) follows the exponential member of the generalized Pareto distribution (Pickands III et al., 1975). The local dimension $d_t$ can then be obtained as the inverse of the distribution's scale

parameter, which can also be expressed as the inverse of the mean of the exceedances:

$$d_t = 1/mean\left(exceedances\right) \tag{A4}$$

The local dimension is an instantaneous metric, and Faranda et al. (2019) have shown that time-averaging of the data can lead to counter-intuitive effects. The most robust approach is therefore to compute $d$ on instantaneous fields. Here, in order to use the same data as for the machine learning, we have used daily means. We have verified that, at least in ERA-Interim, this

has a negligible effect on the average $d$ value for all variables except for geopotential at 100hPa and 200hPa (not shown).

*Author contributions.* SS has conceived the study and methodology, performed the analysis, produced the software and drafted the article. GM contributed to drafting and revising the article.

*Competing interests.* The Authors declare that they have no competing interests

*Acknowledgements.* We thank Peter Dueben for providing the ERA5 data. S. Scher has been funded by the Department of Meteorology (MISU) of Stockholm University. G. Messori was partly supported by the Swedish Research Council Vetenskapsrådet (grant no. 2016-03724). The computations were done on resources provided by the Swedish National Infrastructure for Computing (SNIC) at the High Performance Computing Center North (HPC2N) and National Supercomputer Centre (NSC).

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
