# Peer review of "Weather and climate forecasting with neural networks: using GCMs with different complexity as study-ground"

_Geoscientific Model Development, 2019_

## Referee Comment (RC1) · Peter Düben (Referee) · 21 Mar 2019

This is a very interesting paper that provides an honest presentation of new results on the use of neural networks to learn the equations of motion of the atmosphere. The paper is relevant for GMD and should be published. However, a minor revision that is addressing the comments below could improve the paper.

- As we argue in Dueben and Bauer 2018, I am very surprised that you get away with 1-day timesteps. I would guess that a T21 model with a 1-day timestep would be un-stable when using explicit time-stepping schemes (maybe I am wrong?)

[Figure]

and it is hard to believe that a neural network is learning something like an implicit scheme that would allow for larger timesteps. Can you provide the timestep that is used in Plasim for comparison? If this is much smaller, can you comment why you think that the neural network model may get away with this, in particular towards the pole? (I assume that you are using a regular Gaussian grid where grid-spacing will become smaller towards the pole).

- I am also (positively) surprised that you are having no problems with loss of stability when using neural networks while other papers report problems when using neural networks to represent physical systems. Can you speculate why this is? Maybe due to the convolutional layers? Can you diagnose the change of global energy in climate simulations?

- When using convolutional networks, will the stencil of gridpoints around a gridpoint in the neural network have exactly the same weights for all gridpoints? Or will the weights be changing for each gridpoint? If they are the same, how can you justify that gridpoints towards the poles (that will have a very different resolution for the stencil of surrounding points) use the same weights as for points at the equator?

- The paper would benefit if you would provide more discussion how to improve neural network configurations in future studies at the end of the conclusion (speculations are welcome).

- Figure 4: Why are the PUMA results not symmetric in zonal direction?

Really minor:

Figure 2: The figure should be made bigger to improve visibility.

Page 2: A "local model" as suggested in Dueben and Bauer 2018 would increase the amount of training data significantly since you would train for 40 (or 100) years of data

for N grid points. Maybe worth mentioning? But only if you think this would fit.

Page 2: "...ambitioun has shifted from..." not really. This is rather a different application.

Page 4: "(...see Scher(2018)." add ")"

Page 6: Why zg500 (ta800) and not z500 (t800)?

Page 6: "has higher error at longer lead times" At long lead times and close to zero correlation a bias in the model can cause interesting behaviours in rmse plots. It can also reduce the error.

Page 8: Can you comment how important it was to include the day of year as input?

Figure 5: I do not see the shading in my printout.

Page 11: "to to"

Page 12: Personally, I am very sceptical that the use for different networks for different months would produce reasonable results.

Please feel free to contact me if some of the comments are unclear.

---

## Referee Comment (RC2) · Anonymous Referee #2 · 23 Mar 2019

This paper is a substantial contribution to modelling science within the scope of this journal, containing new concept of producing statistical model for weather forecast by emulating a simplified GCM model, using neural networks. I recommend this paper for publication after minor revision.

Two major points have to be clarifide, otherwise it is difficult to understand the method and results presented in the paper. 1. Naming models. In the paper the same name is used for the GCM and NN emulation of this GCM (e.g., PLASIMT21). This kind of naming create confusion and makes understanding the method and results difficult. 2. What is used as "truth" in Section 2.4 and in Section 3. Are statistics shown in Section
3 represent the accuracy of NN emulations of different models (e.g., NN emulating PLASIMT21 vs. PLASIMT21) or the accuracy of NN emulations vs. reanalysis (e.g., NN emulating PLASIMT21 vs. ERA or NCEP/NCAR reanalysis)?

---

## Referee Comment (RC3) · Vladimir Krasnopolsky (Referee) · 27 Mar 2019

Sebastian, Thank you for clarifications. Looking forward to see your paper published. Best, Vladimir

---

## Short Comment (SC1) · 27 Mar 2019

**Sebastian Scher**

sebastian.scher@misu.su.se

Received and published: 27 March 2019

**Dear Reviewer,**

Thank you for your review and suggestions for improving the clarity of the manuscript. To help other reviewers and readers in the remaining discussion time, I answer your questions already now. Once the discussion is ended, the clarifications will be added to the final manuscript.

"1. Naming models. In the paper the same name is used for the GCM and NN emulation of this GCM (e.g., PLASIMT21). This kind of naming create confusion and makes

understanding the method and results difficult."

Thanks for pointing out to us that this naming created confusion. I agree that the naming convention we use is not ideal, and in the revised manuscript, we will use a naming convention that clearly differentiates between the physical models and the neural network emulations trained on the physical models.

"2.What is used as "truth" in Section 2.4 and in Section 3. Are statistics shown in Section3 represent the accuracy of NN emulations of different models (e.g., NN emulatingPLASIMT21 vs. PLASIMT21) or the accuracy of NN emulations vs. reanalysis (e.g., NN emulating PLASIMT21 vs. ERA or NCEP/NCAR reanalysis"

Throughout the whole paper, the NN emulations are evaluated against the model they were trained on (e.g., the NN trained on PLASIMT21 is evaluated on PLASIMT21). For the "weather forecasting" problem in section 3.1 and 3.2, the networks are tested on forecasting the model a couple of timesteps ahead. This testing is done on the test-set of the modelruns, so the data not used for the training. For the climate statistics in section 3.3, the statistics of the NN climate emulations are compared to the statistics of the model the network has been trained on (e.g. the statistics of the NN trained on PLASIMT21).

In the revised manuscript we will highlight this more to avoid confusion, and make it clearer to the reader that the whole study is confined to the "reality" of the models.

GMDD

---

## Referee Comment (RC4) · Anonymous Referee #3 · 1 Apr 2019

The authors build on a previous paper in which one of the authors applied a convolutional neural net with an encoder-decoder architecture to the problem of weather forecasting and climate simulation in a simplified atmospheric GCM. Here the approach is extended to a more comprehensive atmospheric GCM and to different horizontal resolutions in the GCMs. Inclusion of a seasonal cycle proves to be an important issue when trying to reproduce the climate with the neural net. The research question is exciting and the presentation and approach is generally good. However, changes are needed to properly describe the approach that has been taken and to quantify how well the neural net is doing.
**Major comments**

1. As far as I can tell, the same network as was applied to the simple GCM PUMA is applied to the more comprehensive GCM PLASIM. In particular, the variables used are horizontal winds, geopotential height and temperature. This is surprising since PLASIM presumably has a hydrological cycle, and specific humidity is presumably a prognostic variables. Therefore, the state of the atmosphere in PLASIM at a given time is not described with the 4 variables used. The choice to not include humidity in the network should be justified. Presumably a network with humidity included would do better (?)

2. It is difficult to assess how well the networks are doing in their forecasts in figure 3 because they are not compared to anything else. In the preceding work, comparison was made to 'persistence'. But a more informative choice would be to plot RMSE and ACC for a 'perfect model' forecast in which the same GCM is used to make the forecast with a small perturbation in the initial conditions (or alternatively in the tuned constants in the model physics). Comparing to a perfect model forecast would allow the reader to assess the skill of the neural net forecast - it can't be expected to do better than a perfect model prediction (given any error and the chaotic nature of the atmosphere). This was also help the paper have more impact since the neural net will ultimately have to compete with traditional NWP, albeit in terms of both accuracy and speed and not just accuracy.

3. Another possibility to consider for why the climate prediction with a seasonal cycle does not work well is that you are forecasting over a short time frame (1 day) in which diabatic effects that vary seasonally such as changes in insolation are not very important compared to the dynamical initial condition. Perhaps training using a longer forecast lead time (e.g. 5 days) would work better for the climate simulations with a seasonal cycle.

Minor comments
1. The neural net architecture is described as an autoencoder. My understanding of the nomenclature is that it is an encoder-decoder but not an autoencoder (since the output is not the same as the input).

2. A few more lines description is needed for each GCM in section 2.1. How is the dry dynamical core of PUMA forced? (e.g. is it a Held-Suarez setup?) What makes PLASIM an 'intermediate complexity' GCM? (e.g. how exactly does it differ from a standard GCM aside from the lack of a dynamical ocean).

3. Figure 2 is helpful but not fully described in the caption. In particular the caption should say what the numbers are - does None, 64,128,40 refers to ?, lat, lon, channels. What does 'None' refer to here?

4. Appendix A1: do you use all times t1 before and after t in the calculation?

---

## Short Comment (SC2) · 4 Apr 2019

Dear Reviewer,

Thanks for your comments and suggestions. We will reply in detail once we have received all reviews and prepared a new manuscript. Here I outline how we plan to address your main points and add some clarifications following your minor points.

"1. As far as I can tell, the same network as was applied to the simple GCM PUMA is applied to the more comprehensive GCM PLASIM. In particular, the variables used are horizontal winds, geopotential height and temperature. This is surprising since

PLASIM presumably has a hydrological cycle, and specific humidity is presumably a prognostic variables. Therefore, the state of the atmosphere in PLASIM at a given timeis not described with the 4 variables used. The choice to not include humidity in thenetwork should be justified. Presumably a network with humidity included would dobetter (?)"

You are right, we used the same architecture for PUMA and PLASIM, thus not including moisture, which is indeed part of PLASIM. This choice was made to keep the architecture the same across all models. However, we agree that adding moisture as variable might improve the neural network forecasts trained on PLASIM. In the revised version, we will include a new section where we explore changes/additions to our method. Here we will also include tests with moisture as additional in- and output of our network.

"2. It is difficult to assess how well the networks are doing in their forecasts in figure 3 because they are not compared to anything else. In the preceding work, comparison was made to 'persistence'. But a more informative choice would be to plot RMSE and ACC for a 'perfect model' forecast in which the same GCM is used to make the forecastwith a small perturbation in the initial conditions (or alternatively in the tuned constantsin the model physics). Comparing to a perfect model forecast would allow the reader to assess the skill of the neural net forecast - it can't be expected to do better than aperfect model prediction (given any error and the chaotic nature of the atmosphere).This was also help the paper have more impact since the neural net will ultimately haveto compete with traditional NWP, albeit in terms of both accuracy and speed and no tjust accuracy."

We agree that a baseline forecast would help to aid the reader. In the revised version, we will include the skill of persistence forecasts. We will also explore the possibility of making perfect-model forecasts with PUMA/PLASIM. Additionally, we are planning to include some tests on coarse-grained reanalysis data (as in Dueben and Bauer 2018), which will allow to put our forecast skill in context of the IFS forecasts that Dueben and Bauer used as comparison.

" Another possibility to consider for why the climate prediction with a seasonal cycle does not work well is that you are forecasting over a short time frame (1 day) in which diabatic effects that vary seasonally such as changes in insolation are not very important compared to the dynamical initial condition. Perhaps training using a longer forecast lead time (e.g. 5 days) would work better for the climate simulations with a seasonal cycle"

Thanks for this suggestion. We will add a test with 5-day forecasts in the additional section on changes/additions to out method we plane to include in a revised manuscript.

1. "The neural net architecture is described as an autoencoder. My understanding ofthe nomenclature is that it is an encoder-decoder but not an autoencoder (since theoutput is not the same as the input)."

You are right, our architecture is indeed more correctly described as an encoder-decoder. Thanks for pointing this out, we will change the naming in our revised manuscript

2. "A few more lines description is needed for each GCM in section 2.1. How is the dry dynamical core of PUMA forced? (e.g. is it a Held-Suarez setup?) What makesPLASIM an 'intermediate complexity' GCM? (e.g. how exactly does it differ from a standard GCM aside from the lack of a dynamical ocean)."

PUMA uses Newtonian cooling for diabatic heating/cooling. The main difference between PLASIM and standard GCMs is that all included sub-systems of the Earth system expect for the atmosphere (like the ocean, sea-ice and soil) are reduced to systems of low complexity.

We will include a more detailed description of PUMA and PLASIM in the revised manuscript.

"3. Figure 2 is helpful but not fully described in the caption. In particular the caption should say what the numbers are - does None, 64,128,40 refers to ?, lat, lon, channels.What does 'None' refer to here?"

None, 64,128,40 is the typical notation in Keras and refers to time(or samples), lat, lon, channels. We agree that the notation is confusing to the reader not familiar with the Keras library. We will remove the "None"- that it is not necessary to understand the architecture -in our revised manuscript, and update the caption to better explain the figure.

"4. Appendix A1: do you use all times t1 before and after t in the calculation"

yes, exactly. For computing the local dimension at time t, the distances of the field at t with the fields at all other timesteps are computed.

We will try to make this clearer in the revised manuscript.

---

## Author Comment (AC1) · 15 May 2019

We want to thank all 3 reviewers for their helpful comments that helped to improve our paper. We have now prepared a new version of our manuscript that takes full account of these comments. Specifically, it includes more tests with the neural network setup, and a test on coarse-grained reanalysis data. Additionally, we clarified numerous passages of the text based on the reviewer comments.

Below we list point-by-point responses (in red) to all issues raised by the reviewers. A draft of our new manuscript and a version with track-changes (compared to the original submission) is added at the end of this document.

Reviewer #1

This is a very interesting paper that provides an honest presentation of new results on the use of neural networks to learn the equations of motion of the atmosphere. The paper is relevant for GMD and should be published. However, a minor revision that is addressing the comments below could improve the paper.

• As we argue in Dueben and Bauer 2018, I am very surprised that you get away with 1-day timesteps. I would guess that a T21 model with a 1-day timestep would be unstable when using explicit time-stepping schemes (maybe I am wrong?) and it is hard to believe that a neural network is learning something like an implicit scheme that would allow for larger timesteps. Can you provide the timestep that is used in Plasim for comparison? If this is much smaller, can you comment why you think that the neural network model may get away with this, in particular towards the pole? (I assume that you are using a regular Gaussian grid where grid-spacing will become smaller towards the pole).

We would also expect that a T21 model with a 1-day timestep would be unstable with an explicit time-stepping scheme. However, we do not think that it is appropriate to see a neural network as an explicit time-stepping scheme. Rather, it should be seen as a general mapping function (mapping model states to model states some time later).
Regarding the timestep in the models: we used the standard configuration, which is 30 min for plasimT42, 20min for plasimT21, and 60 min for pumaT21 and pumaT42. We mention this in the new manuscript (p3 L7-8). Regarding the time-step question please see our response to you 2$^{nd}$ point below.
Also, note that we are not using a regular Gaussian grid, but regular lat-lon grids. We now explicitly mention this in the method section.

• I am also (positively) surprised that you are having no problems with loss of stability when using neural networks while other papers report problems when using neural networks to represent physical systems. Can you speculate why this is? Maybe due to the convolutional layers? Can you diagnose the change of global energy in climate simulations?

The reason for not losing stability might indeed be caused by the convolution layers. The reason might be that we use stacked convolution layers, whereas Dueben and Bauer use a local deep network. At a first thought these approaches are very similar, but there are

important differences that might explain why the convolution network works also with longer timesteps. In the simplest setup (1 single convolution layer with stride S and convolution-depth N), the convolution approach is identical to a local network with stride S with 1 single hidden layer with size N. However, as soon as the networks get deeper the differences begin. In the case of adding mode hidden layers to the local network, the additional layers still only get information from stride S. However, when stacking convolution layers, the 2nd convolution layer has more information than from stride S. If S=3, then the second layer already has information from 5 gridpoints, and a 3rd layer from 7 and so on. So what could happen in principle is the following: if 1 single convolution layer can learn let's say a 1 hour forecast, then maybe 24 could learn a 24 hour forecast, whereas each convolution layers in the 24-layer network has exactly the same weights as the single layer network trained on 1 hours. However, we have to note that this a theoretical consideration. We added discussion on this topic in the discussion section (p15 L11-12). Regarding diagnosing the change of global energy in the climate simulations: this would unfortunately be quite difficult to do with our data. Our neural networks are trained on pressure-level data, and therefore also the network-climate simulations are on pressure levels. To our knowledge, computing global atmospheric energy is usually done on model levels, and is very tricky to do on pressure levels. We agree that analyzing global energy would be interesting, but it would require nothing short of an entirely new study with different training for the networks, and we would prefer not to address it here.

• When using convolutional networks, will the stencil of gridpoints around a grid-point in the neural network have exactly the same weights for all gridpoints? Or will the weights be changing for each gridpoint? If they are the same, how can you justify that gridpoints towards the poles (that will have a very different resolution for the stencil of surrounding points) use the same weights as for points at the equator?

In convolution networks the stencil of gridpoints around a grid-point in the neural network has exactly the same weights for all gridpoints (one way to see it is as a moving filter with fixed weights). It is true that the architecture does not take into account that gridpoints towards the poles have different resolution than close to the equator. One way to solve this is to use a spherical formulation of convolution, which has also been proposed on regular grids. We mention this in the new manuscript in the section on potential improvements of the neural network architecture (p.14 L13).

• The paper would benefit if you would provide more discussion how to improve neural network configurations in future studies at the end of the conclusion (speculations are welcome).

We included a new subsection on this at the end of the conclusion section. In this section, we discuss potential network configurations for future studies.

• Figure 4: Why are the PUMA results not symmetric in zonal direction?

The results for PUMA are nearly symmetric when looking at 1-day forecasts (which we didn't show (see fig. R1), but for longer forecasts they are not symmetric any longer. We think this is caused by problems induced at the border of the domain since we did not use any circular wrapping in the convolution. We now mention this in the manuscript (p.9 L1-4).

[Figure]

*Fig. R1: RMSE of network forecasts with leadtime 1 day, for networks trained on PUMAT21(left) and PUMAT42(right).*

Really minor:

Figure 2: The figure should be made bigger to improve visibility.

We increased the size of the figure.

Page 2: A "local model" as suggested in Dueben and Bauer 2018 would increase the amount of training data significantly since you would train for 40 (or 100) years of data for N grid points. Maybe worth mentioning? But only if you think this would fit.

This is in fact also happening with the convolution layers. A convolution layer is a filter of fixed size that is moved across the domain, and the weights are always the same.

Page 2: "...ambition has shifted from..." not really. This is rather a different application.

We changed the sentence to "Recently, in addition to using machine-learning to enhance numerical models, there have been ambitions to use it for weather forecasting itself."

Page 4: "(...see Scher(2018)." add ")"

We made the suggested change

Page 6: Why zg500 (ta800) and not z500 (t800)?

We use this notation because it is the notation used in PUMA and PLASIM. In order to avoid programming errors, we make use of automatic plot-labelling directly form the data.

Page 6: "has higher error at longer lead times" At long lead times and close to zero correlation a bias in the model can cause interesting behaviours in rmse plots. It can also reduce the error.

We added "Here we have to note that at long lead-times and close to zero-correlation, RMSE can be hard to interpret since it can be strongly influenced by biases in the forecasts." (p.7 L24-26)

Page 8: Can you comment how important it was to include the day of year as input?

We have not analyzed the error of multi-day forecasts of the networks that include day of year as input, but the 1-day forecast error is very similar to the error of the standard networks (trained without day of year as input).

Figure 5: I do not see the shading in my printout.

The shading is very narrow (since the spread is so small).

Page 11: "to to"
we removed the double "to".

Page 12: Personally, I am very sceptical that the use for different networks for different months would produce reasonable results.

We agree that it is hard to tell a-priori whether this would work. However, considering the often un-intuitiveness of machine-learning, we prefer to keep this speculative suggestion for further research.

Reviewer #2

This paper is a substantial contribution to modelling science within the scope of this journal, containing new concept of producing statistical model for weather forecast by emulating a simplified GCM model, using neural networks. I recommend this paper for publication after minor revision. Two major points have to be clarified, otherwise it is difficult to understand the method and results presented in the paper.

"1. Naming models. In the paper the same name is used for the GCM and NN emulation of this GCM (e.g., PLASIMT21). This kind of naming create confusion and makes understanding the method and results difficult."
We agree that the naming convention we used was not ideal. In the revised text, we now always speak of "the network for PLASIMT21" or "the PLASIMT21 networks" etc.) when we refer to the neural networks and their forecasts. We hope that this makes the text clearer to the reader.

"2.What is used as "truth" in Section 2.4 and in Section 3. Are statistics shown in Section3 represent the accuracy of NN emulations of different models (e.g., NN emulatingPLASIMT21 vs. PLASIMT21) or the accuracy of NN emulations vs. reanalysis (e.g., NN emulating PLASIMT21 vs. ERA or NCEP/NCAR reanalysis".

Throughout the whole paper, the NN emulations are evaluated against the model they were trained on (e.g., the NN trained on PLASIMT21 is evaluated on PLASIMT21). For the "weather forecasting" problem in sections 3.1 and 3.2, the networks are tested on forecasting the model a couple of timesteps ahead. This testing is done on the test-set of the model runs, so the data not used for the training. For the climate statistics in section 3.3, the statistics of the NN climate emulations are compared to the statistics of the model the network has been trained on (e.g. the statistics of the NN trained on PLASIMT21 are compared to the statistics of PLASIMT21).
We added "All network forecasts will be validated against the underlying model the network was trained on (e.g. for the forecasts of the network trained on PLASIMT21 the ``truth" is the evolution of the PLASIMT21 run)" at the beginning of section 2.4 to make this clearer in the new manuscript (p.6 L20-21).

Anonymous Referee #3

The authors build on a previous paper in which one of the authors applied a convolutional neural net with an encoder-decoder architecture to the problem of weather forecasting and climate simulation in a simplified atmospheric GCM. Here the approach is extended to a more comprehensive atmospheric GCM and to different horizontal resolutions in the GCMs. Inclusion of a seasonal cycle proves to be an important issue when trying to reproduce the climate with the neural net. The research question is exciting and the presentation and approach is generally good. However, changes are needed to properly describe the approach that has been taken and to quantify how well the neural net is doing.

"1. As far as I can tell, the same network as was applied to the simple GCM PUMA is applied to the more comprehensive GCM PLASIM. In particular, the variables used are horizontal winds, geopotential height and temperature. This is surprising since PLASIM presumably has a hydrological cycle, and specific humidity is presumably a prognostic variable. Therefore, the state of the atmosphere in PLASIM at a given time is not described with the 4 variables used.  The choice to not include humidity in the network should be justified.  Presumably a network with humidity included would do better (?)"

The reviewer is correct in noting that we used the same architecture for PUMA and PLASIM, thus not including moisture, which is indeed part of PLASIM. This choice was made to keep the architecture the same across all models. However, we agree that adding moisture as variable might improve the neural network forecasts trained on PLASIM. In the revised version, we have included a test where we include moisture in one of the PLASIM trainings in section 3.5. As you will see, including the 3 hydrological state variables of PLASIM (relative humidity, cloud liquid water content and cloud cover) did not improve the forecasts, but in fact slightly made them worse. This might indicate that our architecture does not – at least not without any changes – work for hydrological variables, which might be related to their very different distributions both in time and space compared to the other variables we use.
Additionally, we now also explicitly mention in the method section that for the main simulations we do not consider the moisture variables in PLASIM in order to avoid confusion (p.6 L6-7).

"2. It is difficult to assess how well the networks are doing in their forecasts in figure 3 because they are not compared to anything else. In the preceding work, comparison was made to 'persistence'. But a more informative choice would be to plot RMSE and ACC for a 'perfect model' forecast in which the same GCM is used to make the forecast with a small perturbation in the initial conditions (or alternatively in the tuned constants in the model physics).  Comparing to a perfect model forecast would allow the reader to assess the skill of the neural net forecast - it can't be expected to do better than a perfect model prediction (given any error and the chaotic nature of the atmosphere). This was also help the paper have more impact since the neural net will ultimately have to compete with traditional NWP, albeit in terms of both accuracy and speed and not just accuracy."

We agree that a baseline forecast would help to aid the reader. In the revised version, we have included the skill of persistence forecasts in fig. 3, and discuss it in the text both in the results and in the conclusion section (p.7, p.15 L1-2).
We also considered making perfect-model forecasts with PUMA and PLASIM as suggested. Unfortunately, this functionality is not implemented per default in PUMA/PLASIM, and to our best knowledge, related software has not been published. Doing perfect model forecasts would require both changes to the restart-routines of the model, and the implementation of an initial perturbation scheme. Especially the latter is not trivial to do. We have therefore chosen to take another approach to put the skill of our network forecasts in a broader context and make them comparable to alternative methods (in addition to including the skill of persistence forecasts). Namely, we included tests on coarse-grained reanalysis data (as in Dueben and Bauer 2018), which is presented in section 3.5 in the new manuscript. For this we used the same subset of ERA5 as in Dueben and Bauer.  Dueben and Bauer have made forecasts on their dataset with the NWP model IFS with T21 resolution. This served as baseline for their neural network forecasts, and can thus also be used to compare our forecasts with via comparing our new Fig. 8 with their Fig. 3a.

3. "Another possibility to consider for why the climate prediction with a seasonal cycle does not work well is that you are forecasting over a short time frame (1 day) in which diabatic effects that vary seasonally such as changes in insolation are not very important compared to the dynamical initial condition. Perhaps training using a longer forecast lead time (e.g. 5 days) would work better for the climate simulations with a seasonal cycle"

We added a test that makes a climate run with a network trained on 5-day forecasts. Unfortunately, this did not resolve the problems with seasonality.
Specifically, we added fig S19 and Video S9, and the following text to the manuscript:

"All our networks were trained to make 1-day forecasts. The influence of the seasonal cycle on atmospheric dynamics - and especially the influence of diabatic effects - may be very small for 1-day predictions. This could make it hard for the network to learn the influence of seasonality. To test this, we repeated the training of the climate network for PLASIMT21 using 5-day forecasts. This made the seasonality of ta800 at a single gridpoint at 76 $^{\circ}$N better (fig. S19 in the supplement), but the fields of the climate run still become unrealistic (video S9 in the supplement)." (p.11 16-20).

Minor comments

1. "The neural net architecture is described as an autoencoder. My understanding of the nomenclature is that it is an encoder-decoder but not an autoencoder (since the output is not the same as the input)."

You are right, our architecture is indeed more correctly described as an encoder-decoder. Thanks for pointing this out, we have changed the naming accordingly in the manuscript.

2. "A few more lines description is needed for each GCM in section 2.1. How is the dry dynamical core of PUMA forced? (e.g. is it a Held-Suarez setup?) What makes PLASIM an 'intermediate complexity' GCM? (e.g. how exactly does it differ from a standard GCM aside from the lack of a dynamical ocean)?"

PUMA uses Newtonian cooling for diabatic heating/cooling. The main difference between PLASIM and standard GCMs is that all included sub-systems of the Earth system except for the atmosphere (like the ocean, sea-ice and soil) are reduced to systems of low complexity.

We now mention these details in the method section (section 2.1)

"3. Figure 2 is helpful but not fully described in the caption. In particular the caption should say what the numbers are - does None, 64,128,40 refers to?, lat, lon, channels. What does 'None' refer to here?"

None, 64,128,40 is the typical notation in Keras and refers to time (or samples), lat, lon, channels. We now explain this in the caption.

"4. Appendix A1: do you use all times t1 before and after t in the calculation"

Yes, exactly. For computing the local dimension at time t, the distances of the field at t with the fields at all other timesteps are computed.

[revised manuscript text omitted]